# Regulatory T Cells but Not Tumour-Infiltrating Lymphocytes Correlate with Tumour Invasion Depth in Basal Cell Carcinoma

**DOI:** 10.3390/diagnostics12122987

**Published:** 2022-11-29

**Authors:** Paranita Ferronika, Safira Alya Dhiyani, Tri Budiarti, Irianiwati Widodo, Hanggoro Tri Rinonce, Sumadi Lukman Anwar

**Affiliations:** 1Department of Anatomical Pathology, Faculty of Medicine, Public Health and Nursing, Dr. Sardjito Hospital, Universitas Gadjah Mada, Yogyakarta 55281, Indonesia; 2Department of Surgery, Faculty of Medicine, Public Health and Nursing, Dr. Sardjito Hospital, Universitas Gadjah Mada, Yogyakarta 55281, Indonesia

**Keywords:** basal cell carcinoma, tumour-infiltrating lymphocyte, regulatory T cell, tumour invasion depth

## Abstract

Basal cell carcinoma (BCC) is the most common skin malignancy worldwide. Current evidence suggests tumour-infiltrating lymphocytes (TILs) may influence the clinical outcomes of patients with BCC. The present study aimed to profile the infiltrative characteristics of stromal TILs and regulatory T cells (Treg cells) in the tumour centre (TC), tumour periphery (TP), and normal adjacent tissue (NAT) of BCC. A total of 111 samples from 43 cutaneous BCC cases were examined for TIL (CD3^+^) and Treg cell (FOXP3^+^/CD3^+^) expression using immunohistochemical techniques. The correlations of Treg cells with TILs, invasion depth, and tumour morphological risk were analysed. We identified a high mean proportion of Treg cells within the tumour (TC = 46.9%, TP = 56.1%, NAT = 51.8%) despite a relatively low median of TILs (TC = 12.7%, TP = 10.3%, NAT = 3.6%), supporting the classification of BCC as a cold tumour. A significant positive correlation was observed between the proportion of Treg cells and sTILs (ρ = 0.325, *p <* 0.001), suggesting a predominant role of TILs in the infiltration of Treg cells. An inverse correlation discovered between Treg cells and tumour invasion depth (r = −0.36, *p =* 0.017) might indicate Treg cells’ anti-tumour capacity in BCC.

## 1. Introduction

Basal cell carcinoma (BCC) is the most diagnosed kind of skin cancer worldwide. The mainstay treatment for BCC is surgical resection with tumour-free excision margins [1]. The discovery of immunotherapy that targets immune-inhibitory checkpoints has shifted the landscape of cancer therapy, particularly for immunogenic cancers, often known as “hot tumours”. Although some studies have reported the success of immunotherapy in BCC, one study reported frequent recurrence in patients with BCC after the administration of anti-PD-1 therapy [2]. Thus, a high recurrence rate is still a major challenge in the management of BCC, regardless of the availability of immunotherapy as the most recent therapeutic advance.

In their case study, Sabbatino et al. proposed that the poor response of BCC to immunotherapy correlates to a low number of tumour-infiltrating lymphocytes (TILs) found in the tumour microenvironment. In contrast to a low number of TILs, the proportion of regulatory T cells (Treg cells)—a subset of CD3^+^ TILs—was reported to be high in BCC [3]. As suggested by some studies, the recruitment of Treg cells to tumour sites inhibits the activation and differentiation of other CD3^+^ TILs, subsequently decreasing the host’s immune response to tumourigenesis and invasive processes [4]. The pro-tumour effect of Treg cells was supported by evidence showing the correlation between increased Treg cell proportion with a heightened recurrence rate found in patients with non-small-cell lung cancer [5] and hepatocellular carcinoma [6]. However, in other studies, a high density of Treg cells in the tumours was found to be associated with increased CD8+ cytotoxic T lymphocytes (CTLs) capable of the direct elimination of cancer cells [7,8]. The activation of CD8^+^ CTLs by Treg cells might explain the correlation between the high number of Treg cells with lower recurrence rates in patients with estrogen receptor-negative (ER^-^) breast cancer [7,8], clear cell renal cell carcinoma [9], and bladder cancer [10]. The controversy over the effects of Treg cells among different tumour types and their correlation with clinical outcomes emphasise the importance of evaluating Treg cell infiltration characteristics in each specific tumour type. The presence of Treg cells in each type of tumour is also important to be further studied due to its potential as a novel therapeutic approach.

The present study aimed to evaluate the proportion of stromal TILs (sTILs) and Treg cells to profile their infiltration characteristics among different tumour zones within each patient with BCC. As recommended by the International Immuno-Oncology Biomarker Working Group [11], CD3^+^ sTILs were evaluated in the tumour centre, tumour periphery, and the normal adjacent tissue beyond the tumour borders. Similar to sTILs, CD3^+^/FOXP3^+^ Treg cells were also evaluated in the three different tumour areas. We also analysed the correlations of Treg cell proportions with tumour morphological risk and tumour invasion depth in our samples. The evaluation of sTIL and Treg cell infiltration patterns, and also their correlations with tumour characteristics hopefully gives better insights into the role of sTILs and Treg cells in the development and progression of BCC. 

## 2. Materials and Methods

### 2.1. Sample Specifications and Collection

During the years 2016–2021, 132 BCC cases were identified in the Anatomical Pathology Installation of Dr. Sardjito Hospital, Yogyakarta, Indonesia. Among 132 cases, 69 cases were classified as cutaneous BCC. Of 69 cases, 26 were excluded because of the absence of both normal adjacent tissue and tumour periphery in their respective paraffin blocks; hence, they were unable to be used for comparative analyses, leaving 43 cases to be included in our study. In total, 111 slides from 3 different tumour zones were acquired for the subsequent immunohistochemical assessment: 43 from the tumour centre (TC), 34 from the tumour periphery (TP), and 34 from the normal adjacent tissue (NAT). TP is defined as the zone centred on the tumour border separating the normal adjacent tissue from the tumour nests with an extent of 2 mm. TC is the zone next to TP toward the centre of the tumour, while NAT is the dermal zone beyond the TP. 

### 2.2. Immunohistochemistry (IHC): FOXP3/CD3 Double Staining

Immunostaining procedures were conducted in the Anatomical Pathology Installation of Dr. Sardjito Hospital, Yogyakarta, Indonesia. Ultratek HRP Anti-polyvalent (DAB) staining system kit (ScyTek Laboratories Inc., West Logan, UT, USA) was utilised to perform deparaffinisation, rehydration, blocking of endogenous peroxidase, antigen retrieval, and counterstaining. AEC substrate chromogen was purchased separately from the staining system kit. The antibodies used for double staining were FOXP3 mouse monoclonal antibody (236A/E7, Abcam, Cambridge, UK) diluted to a 1:400 ratio and CD3 mouse monoclonal antibody (Novocastra Laboratories, Newcastle upon Tyne, UK) diluted to a 1:100 ratio. Hematoxylin and Bluing Reagent were used to counterstain the slides. Normal spleen tissue was used as a positive control.

### 2.3. Immunostaining and Pathological Parameter Assessment

Immunostained IHC slides were examined for FOXP3 and CD3 expressions by two independent observers. Each slide was viewed at 400× magnification and 5 fields were selected to be digitally captured using the OptiLab Viewer (PT. Miconos, Sleman, Indonesia). The captured images were manually counted for regulatory T cells (Treg cells) defined as FOXP3/CD3 double-positive stained cells (brown nucleus and red cell membrane), and sTILs defined as CD3 single-positive cells (red cell membrane) with the aid of an on-screen count tracker in the software ImageJ (NIH, Madison, WI, USA). Treg cell proportion was defined as the fraction of Treg cell count (double-positive cells) over the sum of Treg cells and non-Treg cell lymphocytes (the sum of double-positive and single-positive cells). Meanwhile, the sTIL proportion was defined as the area occupied by all CD3^+^ cells over the area of the stroma. Tumour invasion depth was assessed at the thickest area at the centre of the tumour. Tumour morphological risk was classified as low-risk and high-risk based on the latest version of the World Health Organization (WHO) classification of skin tumours [12]. 

### 2.4. Statistical Analysis

Statistical analyses were performed using SPSS 26.0 (IBM Corp., Armonk, NY, USA). The difference in sTIL and Treg cell counts amongst three different tumour zones (TC, TP, and NAT) were analysed using the Kruskal–Wallis test. Meanwhile, the difference in Treg cell proportions amongst the three different tumour zones was measured using one-way ANOVA. The correlation of sTILs with tumour invasion depth was analysed using Spearman’s rank correlation; meanwhile, the correlation of Treg cell proportion with tumour invasion depth was investigated using Pearson’s correlation coefficient. The difference in sTIL and Treg cell proportion between low and high-morphological-risk groups were calculated using Mann–Whitney U test and independent *t*-test, respectively. A p value of <0.05 was considered statistically significant.

## 3. Results

A total of 111 samples from 43 cutaneous BCC cases were evaluated for sTILs, Treg cell counts, and Treg cell proportions. Samples were divided into three different tumour zones: tumour centre (TC), tumour periphery (TP), and normal adjacent tissue (NAT) (Figure 1 and Figure 2). Minimal lymphocytic infiltrates were found in our samples, with a cumulative median sTIL count of 9%. The TC was found to have the highest sTIL median density (12.7%) (Figure 3a and Table 1), followed by the TP and NAT (10.3% and 3.6%, respectively). A significant difference in sTILs between the groups was observed (*p* = 0.002). A post hoc analysis confirmed a significant difference in sTILs between the TP and NAT (*p* = 0.012), as well as between the TC and NAT (*p* = 0.004). No significant difference in sTILs between the TC and TP was found.

The median Treg cell count was observed to be highest in the TP with 11 cells, followed by the TC with seven cells, and lowest in the NAT with five cells (Figure 3b and Table 1). Statistical analysis showed a significant difference in the Treg cell counts’ median among tumour zones (*p* = 0.014), with post hoc analysis affirming the disparity between the counts in the TP and NAT to be most prominent (*p* = 0.01). However, the difference in Treg cell proportion among the three tumour zones was found to be statistically insignificant (*p* = 0.195), with a mean of 46.9% in the TC, 56.1% in the TP, and 51.8% in the NAT (Figure 3c and Table 1). To observe the interaction between sTILs and Treg cells, we conducted bivariate analysis and found a significant correlation between both parameters, in which the increment of sTILs was always followed by the increase in Treg cell proportion (ρ = 0.325, *p <* 0.001) (Figure 3d and Table 1).

To explore the effects of Treg cell infiltrates on tumour characteristics, we analysed the association of Treg cell proportion with tumour invasion depth and morphological risk. A significant negative correlation was found between tumour invasion depth and Treg cell proportion in the TC (r = −0.36, *p =* 0.017), but not in the TP (r = −0.029, *p =* 0.873) (Figure 3e and Table 1). Inconsistent results of the correlations between Treg cell proportion and tumour invasion depth among different tumour zones highlight the necessity of a specific region sampling for the Treg cell infiltration assessment. No significant difference of Treg cell proportion between low and high-morphological-risk groups was found within each tumour zone (*p* = 0.676 in TC; and *p* = 0.957 in TP) (Figure 3f and Table 1).

## 4. Discussion

In the present study, the tumour microenvironment (TME) of our BCC samples demonstrated a high proportion of tumour-associated Treg cells, despite a relatively low proportion of sTILs. These results suggest BCC cases in our study belong to the category of phenotypically “cold” tumours. Cold tumours are malignancies characterised by the dominance of the Treg cell proportion over the amount of all of the other sTILs, such as natural killer (NK) cells, CD8^+^ T cells, and T helper 1 (T_H_1) cells in the TME [13]. In contrast, NK cells, CD8^+^ T cells, and T_H_1 cells are found in high numbers among phenotypically hot tumours [13]. The classification of BCC as a cold tumour is supported by another study showing a high stromal FoxP3^+^ to CD4^+^ T cell proportion, with a mean of 45%, concordant with our finding of 51.3% [14]. Other malignancies also classified as cold tumours include pancreatic cancer, neuroblastoma, and prostate cancer [15]. Meanwhile, the hot tumour group is comprised of melanoma, non-small-cell lung cancer, Merkel cell carcinoma, and head and neck squamous cell carcinoma (HNSCC) [15]. In the clinical setting, the difference between the two groups was noted mainly for their responses to immunotherapy, particularly to immune checkpoint inhibitors (ICIs). Malignancies in the hot tumour group have generally shown a better outcome in response to ICIs, such as anti-PD-1, anti-PD-L1, and anti-CTLA-4 drugs; thus, an alternative therapeutic strategy for cold tumours needs to be explored further [16].

Even though the quantification of TILs has been included as part of a routine diagnostic procedure in pathology [11], the individual components of TILs were only recently suggested to better correlate with the clinical outcomes of patients [17,18]. TILs are composed of a heterogeneous group of cells, including dendritic cells (DCs), natural killer (NK) cells, myeloid-derived suppressor cells (MDSCs), macrophages, and, most importantly, T cells [19]. The most common T lymphocytes are CD3^+^, CD4^+^ (T_H_1 and T_H_2 cells), CD8^+^, and FoxP3^+^ Treg cells. Even though the presence of Treg cells in tumours has been suggested to affect clinical outcomes, controversy regarding whether it is the accretion [20,21] or depletion of Treg cells [22] that correlates with longer overall survival (OS) is still ongoing. The inconsistency of results on the association between Treg cells and OS may implicate the heterogenous nature of Treg cells and, consequently, their divergent influences on various TIL components in various tumour subtypes [4,7,8].

Aside from tumour subtypes, the tumour zones in which Treg cells were examined seem to contribute to inconsistent results among different studies. In the present study, we found significantly different Treg cell counts between the TC, TP, and NAT. Interestingly, in alignment with our findings, the number of Treg cells was found to be significantly higher within the tumour sites of BCC [3,23], breast cancers [24], pancreatic cancers [25], and colorectal cancers [26] compared to their matched NATs, regardless of their association with the status of lymph node or distant metastasis. The tumour phenotypes, chemokine expression, oxidative status, metabolism, and pH in TME were suggested to influence the infiltration of Treg cells to tumour sites [25].

Within the tumour, we found higher sTIL and Treg cell proportions in the TP than in the TC. This phenomenon might be explained by the different main activities between these tumour zones. The pathways activated in the TC are mainly related to altered cellular metabolisms, which include tumour cell adaptation to hypoxia and necrosis [27]. On the other hand, the pathways activated in the TP are mainly linked to stress and inflammation, which include NF-kappa B signalling, Toll-like receptor signalling, and NOD-like receptor signalling. NF-kappa B signalling is important in BCC development through the activation of the STAT3 signalling pathway [28]. Other inflammatory cytokines involved in the JAK-STAT pathway, such as EGF, IL-6, and Cxc11, were also involved in BCC development. Interactions among the components of inflammatory and tumour cells add to the complexity of the TME. Thus, considerations towards the presence of spatial heterogeneity of inflammatory cells is crucial to optimally portray the TME of BCC. 

In our study, we discovered a significant positive correlation between the proportion of Treg cells and sTILs in all three observed zones. We also did not find the increase in Treg cell proportion in the TC and/or TP compared to the NAT to be significant. It can be inferred from these results that the infiltration of Treg cells into the TME seems to be largely regulated by sTILs, rather than independently by the tumour cells. Previous studies have shown that the recruitment, expansion, and conversion processes of peripheral Treg cells into active cells involve various chemokine–chemokine receptor interactions regulated predominantly by cytokines released by tumour-associated macrophages (TAM) and partly by NK cells [29,30]. The activities of TAMs which support the maintenance and enrichment of the Treg cell population are carried out through modulator cytokines released by T_H_1 and T_H_2 cells [31]. In response, TAMs release cytokines that induce the activation of T_H_1 and T_H_2 cells, resulting in a continuous cycle of TIL–TAM upregulation, which enhances the density of Treg cells within the TME [32]. 

With respect to tumour characteristics, we found a significant correlation of Treg cell proportion with the tumour invasion depth. Tumour invasion depth is largely accepted as an independent prognostic factor in skin melanoma [33]. Albeit the sparsity of recent investigations, tumour depth in BCC is significantly associated with tumour aggressiveness, indicated by the presence of squamous metaplasia or a micronodular pattern, perineural invasion, poor differentiation, and high collagen composition [34,35]. Another study also reported a 3.1 times higher odds ratio for local recurrence and metastasis or death when the tumour extends beyond the subcutaneous adipose tissue [36]. In the present study, we discovered higher tumour invasion depth correlates with a lower Treg cell proportion in the TC. Whereas the pro-tumour role of Treg cells was largely accepted, the negative correlation between tumour invasion depth and Treg cells in the present study seems to support the anti-tumour capacity of Treg cells. The anti-tumour role of Treg cells might be facilitated by CTLs’ immune response induced by the presence of Treg cells in certain tumour types [7,8]. In tumour cells, the presence of CTLs is very important in inhibiting tumour progression by releasing cytotoxic molecules, as well as inducing tumour cell apoptosis [13]. The anti-tumour role of Treg cells and CTLs is supported by previous studies which showed the increment in Treg cells [20,21] and CTLs in some tumour types [37,38,39] correlates with a longer OS. Unfortunately, the assessment of CTL infiltration was not included in our study. The significant correlation between tumour invasion depth and Treg cells being only found in the TC might be affected by the sample characteristics, in which the deepest tumour invasion was mainly measured in the TC.

We found no significant difference in sTIL and Treg cell proportion between different tumour morphological risk groups. The morphological risk group stratifies BCC into two categories (low and high-risk) based on the risk of recurrence [40]. The low-risk group includes nodular, superficial, pigmented, infundibulocystic, and fibroepithelial subtypes. Meanwhile, the high-risk group includes basosquamous, morphoeic, infiltrating, micronodular, and BCC with sarcomatoid differentiation [12]. A limited amount of the literature has studied the association of these morphological subtypes with inflammatory cells infiltrating the TME. A previous study showed that inflammatory-induced cytokines, CXCR4, MMP-13, and β-catenin, are highly expressed in the basosquamous subtype, moderately expressed in the superficial subtype, and unremarkably expressed in the micronodular subtype [41]. Thus, a two-tier classification of morphological risk seems to inadequately represent the association between BCC subtypes and inflammatory reactions surrounding the tumour. Further studies with a higher sample size for each morphological subtype is needed to analyse the distribution of inflammatory cells and cytokines among different morphological subtypes of BCC. 

In summary, the present research findings imply that the diagnostic approach through the quantification of specific components of TILs rather than TILs by themselves seems to pose a greater benefit for the determination of patient management and prognosis. The selection strategy for tumour zone sampling needs to be appraised considering the heterogenous distribution of TILs and their components surrounding the tumour. The correlation between tumour invasion depth, Treg cells, and potentially patient outcomes in BCC further enhances the benefits of Treg cell quantification in clinical practice. The strength of our study lies in the use of multiple tumour zones to analyse the variations of sTIL and Treg cell distribution within different tumour zones. However, this multiple-tumour-zone sampling method limits our sample selection to only BCC cases with paraffin blocks containing tissue from the tumour periphery and adjacent tumour-free tissue. From the 69 cutaneous BCC cases found in our hospital, 23 cases were excluded because tissues containing the tumour periphery and tumour-free area were absent in their respective paraffin blocks. Due to the sampling method, our cases are more representative of incompletely resected BCC cases with a relatively higher invasion depth (mean = 6.4 mm) than the general BCC cases.

## Figures and Tables

**Figure 1 diagnostics-12-02987-f001:**
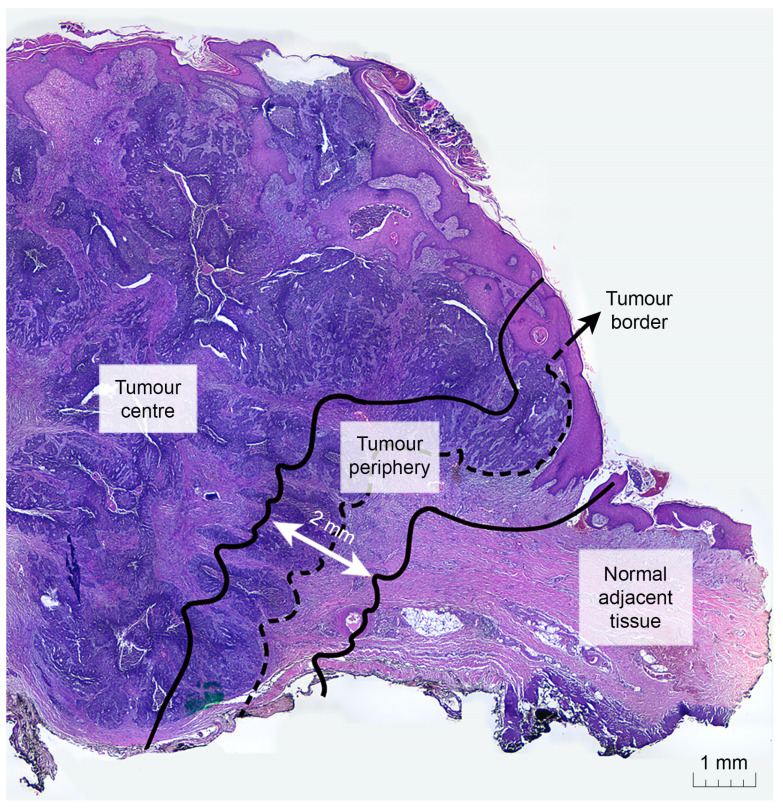
The tumour periphery (TP) is defined as the zone centred on the tumour border separating the normal adjacent tissue (NAT) from the tumour nests, with an extent of 2 mm. The tumour centre (TC) corresponds to all the tissue on the inner side of the TP, while NAT refers to the area on the outer side of the TP.

**Figure 2 diagnostics-12-02987-f002:**
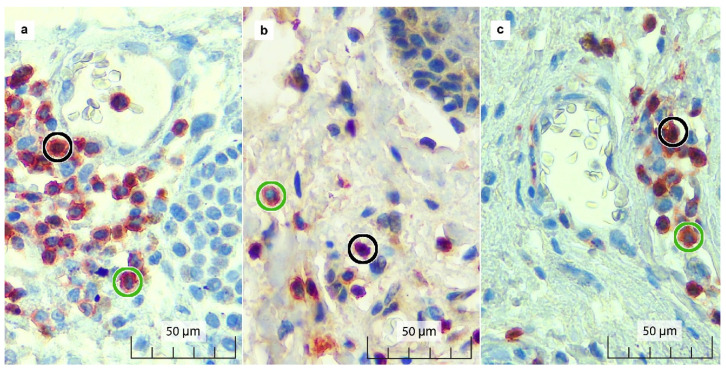
Double immunostaining results for FOXP3 and CD3 in the (**a**) TC, (**b**) TP, and (**c**) NAT. FOXP3+/CD3+ (double-positive) Treg cells show brown-stained nuclei surrounded by red-stained cell membranes (black circle). CD3+ (single-positive) sTILs show red-stained cell membranes with unstained blue nuclei (green circle).

**Figure 3 diagnostics-12-02987-f003:**
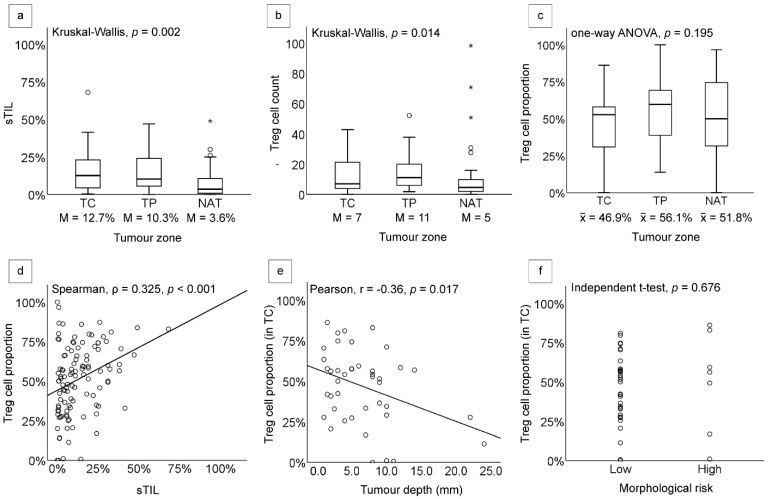
Boxplots and scatterplots showing the associations between observed variables in tumour centre (TC), tumour periphery (TP), and the normal adjacent tissue (NAT): (**a**) The proportion of sTILs was highest in the TC, and a significant difference was found between the sTIL medians of NAT and TP (*p* = 0.012), as well as between NAT and TC (*p* = 0.004); (**b**) The difference in Treg cell counts was statistically significant between NAT and TP (*p* = 0.009), as well as between NAT and TC (*p* = 0.018); (**c**) Treg cell proportions between the three tumour zones showed no significant difference; (**d**) Significant positive correlation was observed between sTIL and Treg cell proportion (ρ = 0.325, *p* < 0.001); (**e**) Pearson correlation showed a significant negative correlation between tumour invasion depth and Treg cell proportion in the TC (r = −0.362, *p* = 0.017); (**f**) No significant difference of Treg cell proportion was observed between morphological risk in the TC (*p* = 0.676). In the boxplots (**a**,**b**), circles represent potential outliers while * (stars) represent true outliers. x̅ = mean, M = median.

**Table 1 diagnostics-12-02987-t001:** Summary of the associations between observed parameters.

		sTILs		Treg Cell Count		Treg Cell Proportion	
		Median (%)	*p* Value	Median (Cell)	*p* Value	Mean (%)	*p* Value
Tumour zone			0.002 (Kruskal–Wallis)		0.014 (Kruskal–Wallis)		0.195 (ANOVA)
	TC	12.7		7		46.9	
	TP	10.3		11		56.1	
	NAT	3.6		5		51.8	
Tumour depth *			0.076 (Spearman’s ρ)		0.027 (Spearman’s ρ)		0.017 (Pearson’s r)
Morphological risk **							
	Low	11.1	0.573 (Mann–Whitney)	9	0.392 (Mann–Whitney)	46.3	0.676 (Independent *t*-test)
	High	19.0		6		50.2	

TC: tumour centre; TP: tumour periphery; NAT: normal adjacent tissue. * No categorical data for tumour depth; only *p*-value is provided. ** The difference of dependent variables between two morphological risk groups measured in TC.

## Data Availability

All data that contributed to the conclusions of this study are included in this article. Further enquiries can be directed to the corresponding author.

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
