# Peer review of "Regulatory T Cells but Not Tumour-Infiltrating Lymphocytes Correlate with Tumour Invasion Depth in Basal Cell Carcinoma"

_diagnostics, 2022, doi:10.3390/diagnostics12122987_

Round 1
Reviewer 1 Report
Dear authors,
the paper you submitted is of high quality, interesting and results are presented clearly.
I have three minor suggestions to improve the article:
1. when you describe sample you stated having 43 of 69 samples due to adequate samples. Could you add a line or two on 26 samples that were not adequate for analysis
2. in statistical analysss you mentioned using Mann-Whitney U test to assess correlations. Mann-Whitney U test detect if there is a significant difference between two groups so please correct it (We used Mann-Whitney U test to assess difference between groups of samples)
3. in discussion, the paragraph where you comment the limitations of the study please add comments on 26 samples you could not use (could they differ from samples in the study and could it possibly distort your results and interpretation). Furthermore, add a line or two on representativity of your sample (are the samples from your hospital representative for all BCC patients, or maybe you have patients with more/less severe cases comparing to other hospitals...)
Thank you for taking these notes into consideration
Author Response
Dear reviewer,
We are grateful to have received suggestions from you and herewith applied the recommendations to improvise our manuscript. Track changes have been included in our manuscript as instructed in the revision manual.
- when you describe sample you stated having 43 of 69 samples due to adequate samples. Could you add a line or two on 26 samples that were not adequate for analysis
Comment: we have revised the explanation of the reason for exclusion in a more detailed manner (lines 70-73).
- in statistical analysss you mentioned using Mann-Whitney U test to assess correlations. Mann-Whitney U test detect if there is a significant difference between two groups so please correct it (We used Mann-Whitney U test to assess difference between groups of samples)
Comment: we have corrected all the inferences from Mann-Whitney U tests in the manuscript from “correlation” to “difference” (lines 112-116, 162-164, 170-171, & 184)
- in discussion, the paragraph where you comment the limitations of the study please add comments on 26 samples you could not use (could they differ from samples in the study and could it possibly distort your results and interpretation). Furthermore, add a line or two on representativity of your sample (are the samples from your hospital representative for all BCC patients, or maybe you have patients with more/less severe cases comparing to other hospitals...)
Comment: we have added several lines to elaborate on the characteristics of our samples which may set our study apart from similar studies (lines 294-303).
Additionally, we have also updated the file for figure 1 as there was a minor error in the previous picture (disconnect of the arrow at the end of the dashed line).
We truly appreciate all the constructive comments and suggestions and hope that the changes made properly resolved all the concerns addressed.
Best regards,
Authors
Reviewer 2 Report
Dear authors,
It is a well-designed article, I think it will contribute to the literature. Here are some of my editing suggestions:
- There are grammatical errors in English, I suggest they be corrected.
- I would expect the discussion section to be more comprehensive in such a comprehensive study.
Best wishes....
Author Response
Dear Reviewer,
We are grateful to have received suggestions from you and herewith applied the recommendations to improvise our manuscript. Track changes have been included in our manuscript as instructed in the revision manual.
- There are grammatical errors in English, I suggest they be corrected.
Comment: we have reviewed our manuscript and made several changes in the text related to grammatical errors. The edited phrases have been highlighted by track changes.
- I would expect the discussion section to be more comprehensive in such a comprehensive study.
Comment: we have added two more paragraphs in the discussion section, to elaborate and compare our findings with previous studies on the significance of studying the difference of inflammatory activities in the TME among different tumour zones, the pathways involved in those activities, the difference of Treg and TILs distribution between morphological risk groups, and the significance of our sample characteristics related to our sampling method (lines 226-237, 273-287).
Additionally, we have also updated the file for figure 1 as there was a minor error in the previous picture (disconnect of the arrow at the end of the dashed line).
We truly appreciate all the constructive comments and suggestions and hope that the changes made properly resolved all the concerns addressed.
Best regards,
Authors